# Online convex optimization for cumulative constraints

**Jianjun Yuan**
Department of Electrical and Computer Engineering
University of Minnesota
Minneapolis, MN, 55455
yuanx270@umn.edu

**Andrew Lamperski**
Department of Electrical and Computer Engineering
University of Minnesota
Minneapolis, MN, 55455
alampers@umn.edu

## Abstract

We propose the algorithms for online convex optimization which lead to cumulative squared constraint violations of the form $\sum_{t=1}^{T} \left([g(x_t)]_+\right)^2 = O(T^{1-\beta})$, where $\beta \in (0,1)$. Previous literature has focused on long-term constraints of the form $\sum_{t=1}^{T} g(x_t)$. There, strictly feasible solutions can cancel out the effects of violated constraints. In contrast, the new form heavily penalizes large constraint violations and cancellation effects cannot occur. Furthermore, useful bounds on the single step constraint violation $[g(x_t)]_+$ are derived. For convex objectives, our regret bounds generalize existing bounds, and for strongly convex objectives we give improved regret bounds. In numerical experiments, we show that our algorithm closely follows the constraint boundary leading to low cumulative violation.

## 1 Introduction

Online optimization is a popular framework for machine learning, with applications such as dictionary learning [14], auctions [1], classification, and regression [3]. It has also been influential in the development of algorithms in deep learning such as convolutional neural networks [11], deep Q-networks [15], and reinforcement learning [8, 20].

The general formulation for online convex optimization (OCO) is as follows: At each time t, we choose a vector $x_t$ in convex set $S = \{x : g(x) \le 0\}$. Then we receive a loss function $f_t : S \to R$ drawn from a family of convex functions and we obtain the loss $f_t(x_t)$. In this general setting, there is no constraint on how the sequence of loss functions $f_t$ is generated. See [21] for more details.

The goal is to generate a sequence of $x_t \in S$ for $t = 1, 2, .., T$ to minimize the cumulative regret which is defined by:

$$Regret_T(x^*) = \sum_{t=1}^{T} f_t(x_t) - \sum_{t=1}^{T} f_t(x^*) \tag{1}$$

where $x^*$ is the optimal solution to the following problem: $\min_{x \in S} \sum_{t=1}^{T} f_t(x)$. According to [2], the solution to Problem (1) is called Hannan consistent if $Regret_T(x^*)$ is sublinear in $T$.

For online convex optimization with constraints, a projection operator is typically applied to the updated variables in order to make them feasible at each time step [21, 6, 7]. However, when the constraints are complex, the computational burden of the projection may be too high for online computation. To circumvent this dilemma, [13] proposed an algorithm which approximates the true desired projection with a simpler closed-form projection. The algorithm gives a cumulative regret $Regret_T(x^*)$ which is upper bounded by $O(\sqrt{T})$, but the constraint $g(x_t) \leq 0$ may not be satisfied in every time step. Instead, the long-term constraint violation satisfies $\sum_{t=1}^{T} g(x_t) \leq O(T^{3/4})$, which is useful when we only require the constraint violation to be non-positive on average: $\lim_{T \to \infty} \sum_{t=1}^{T} g(x_t)/T \leq 0$.

More recently, [10] proposed an adaptive stepsize version of this algorithm which can make $Regret_T(x^*) \leq O(T^{\max\{\beta, 1-\beta\}})$ and $\sum_{t=1}^{T} g(x_t) \leq O(T^{1-\beta/2})$. Here $\beta \in (0, 1)$ is a user-determined trade-off parameter. In related work, [19] provides another algorithm which achieves $O(\sqrt{T})$ regret and a bound of $O(\sqrt{T})$ on the long-term constraint violation.

In this paper, we propose two algorithms for the following two different cases:

**Convex Case:** The first algorithm is for the convex case, which also has the user-determined trade-off as in [10], while the constraint violation is more strict. Specifically, we have $Regret_T(x^*) \leq O(T^{\max\{\beta, 1-\beta\}})$ and $\sum_{t=1}^{T} ([g(x_t)]_+)^2 \leq O(T^{1-\beta})$ where $[g(x_t)]_+ = \max\{0, g(x_t)\}$ and $\beta \in (0, 1)$. Note the square term heavily penalizes large constraint violations and constraint violations from one step cannot be canceled out by strictly feasible steps. Additionally, we give a bound on the cumulative constraint violation $\sum_{t=1}^{T} [g(x_t)]_+ \leq O(T^{1-\beta/2})$, which generalizes the bounds from [13, 10].

In the case of $\beta = 0.5$, which we call "balanced", both $Regret_T(x^*)$ and $\sum_{t=1}^{T} ([g(x_t)]_+)^2$ have the same upper bound of $O(\sqrt{T})$. More importantly, our algorithm guarantees that at each time step, the clipped constraint term $[g(x_t)]_+$ is upper bounded by $O(\frac{1}{T^{1/6}})$, which does not follow from the results of [13, 10]. However, our results currently cannot generalize those of [19], which has $\sum_{t=1}^{T} g(x_t) \leq O(\sqrt{T})$. As discussed below, it is unclear how to extend the work of [19] to the clipped constraints, $[g(x_t)]_+$.

**Strongly Convex Case:** Our second algorithm for strongly convex function $f_t(x)$ gives us the improved upper bounds compared with the previous work in [10]. Specifically, we have $Regret_T(x^*) \leq O(\log(T))$, and $\sum_{t=1}^{T} [g(x_t)]_+ \leq O(\sqrt{\log(T)T})$. The improved bounds match the regret order of standard OCO from [9], while maintaining a constraint violation of reasonable order.

We show numerical experiments on three problems. A toy example is used to compare trajectories of our algorithm with those of [10, 13], and we see that our algorithm tightly follows the constraints. The algorithms are also compared on a doubly-stochastic matrix approximation problem [10] and an economic dispatch problem from power systems. In these, our algorithms lead to reasonable objective regret and low cumulative constraint violation.

## 2 Problem Formulation

The basic projected gradient algorithm for Problem (1) was defined in [21]. At each step, $t$, the algorithm takes a gradient step with respect to $f_t$ and then projects onto the feasible set. With some assumptions on $S$ and $f_t$, this algorithm achieves a regret of $O(\sqrt{T})$.

Although the algorithm is simple, it needs to solve a constrained optimization problem at every time step, which might be too time-consuming for online implementation when the constraints are

complex. Specifically, in [21], at each iteration $t$, the update rule is:

$$x_{t+1} \quad = \Pi_S(x_t - \eta \nabla f_t(x_t)) \quad = \arg\min_{y \in S} \|y - (x_t - \eta \nabla f_t(x_t))\|^2 \qquad (2)$$

where $\Pi_S$ is the projection operation to the set $S$ and $\| \quad \|$ is the $\ell_2$ norm.

In order to lower the computational complexity and accelerate the online processing speed, the work of [13] avoids the convex optimization by projecting the variable to a fixed ball $S \subseteq \mathcal{B}$, which always has a closed-form solution. That paper gives an online solution for the following problem:

$$\min_{x_1,\dots,x_T \in \mathcal{B}} \quad \sum_{t=1}^T f_t(x_t) - \min_{x \in S} \sum_{t=1}^T f_t(x) \quad s.t. \quad \sum_{t=1}^T g_i(x_t) \leq 0, i = 1, 2, \dots, m \qquad (3)$$

where $S = \{x : g_i(x) \leq 0, i = 1, 2, \dots, m\} \subseteq \mathcal{B}$. It is assumed that there exist constants $R > 0$ and $r < 1$ such that $r\mathbb{K} \subseteq S \subseteq R\mathbb{K}$ with $\mathbb{K}$ being the unit $\ell_2$ ball centered at the origin and $\mathcal{B} = R\mathbb{K}$.

Compared to Problem (1), which requires that $x_t \in S$ for all $t$, (3) implies that only the sum of constraints is required. This sum of constraints is known as the *long-term constraint*.

To solve this new problem, [13] considers the following augmented Lagrangian function at each iteration $t$:

$$\mathcal{L}_t(x, \lambda) = f_t(x) + \sum_{i=1}^m \left\{ \lambda_i g_i(x) - \frac{\sigma\eta}{2} \lambda_i^2 \right\} \qquad (4)$$

The update rule is as follows:

$$x_{t+1} = \Pi_\mathcal{B}(x_t - \eta \nabla_x \mathcal{L}_t(x_t, \lambda_t)), \quad \lambda_{t+1} = \Pi_{[0,+\infty)^m}(\lambda_t + \eta \nabla_\lambda \mathcal{L}_t(x_t, \lambda_t)) \qquad (5)$$

where $\eta$ and $\sigma$ are the pre-determined stepsize and some constant, respectively.

More recently, an adaptive version was developed in [10], which has a user-defined trade-off parameter. The algorithm proposed by [10] utilizes two different stepsize sequences to update $x$ and $\lambda$, respectively, instead of using a single stepsize $\eta$.

In both algorithms of [13] and [10], the bound for the violation of the long-term constraint is that $\forall i$, $\sum_{t=1}^T g_i(x_t) \leq O(T^\gamma)$ for some $\gamma \in (0, 1)$. However, as argued in the last section, this bound does not enforce that the violation of the constraint $x_t \in S$ gets small. A situation can arise in which strictly satisfied constraints at one time step can cancel out violations of the constraints at other time steps. This problem can be rectified by considering clipped constraint, $[g_i(x_t)]_+$, in place of $g_i(x_t)$.

For convex problems, our goal is to bound the term $\sum_{t=1}^T \left([g_i(x_t)]_+\right)^2$, which, as discussed in the previous section, is more useful for enforcing small constraint violations, and also recovers the existing bounds for both $\sum_{t=1}^T [g_i(x_t)]_+$ and $\sum_{t=1}^T g_i(x_t)$. For strongly convex problems, we also show the improvement on the upper bounds compared to the results in [10].

In sum, in this paper, we want to solve the following problem for the general convex condition:

$$\min_{x_1,x_2,\dots,x_T \in \mathcal{B}} \quad \sum_{t=1}^T f_t(x_t) - \min_{x \in S} \sum_{t=1}^T f_t(x) \qquad s.t. \quad \sum_{t=1}^T \left([g_i(x_t)]_+\right)^2 \leq O(T^\gamma), \forall i \qquad (6)$$

where $\gamma \in (0, 1)$. The new constraint from (6) is called the *square-clipped long-term constraint* (since it is a square-clipped version of the long-term constraint) or *square-cumulative constraint* (since it encodes the square-cumulative violation of the constraints).

To solve Problem (6), we change the augmented Lagrangian function $\mathcal{L}_t$ as follows:

$$\mathcal{L}_t(x, \lambda) = f_t(x) + \sum_{i=1}^m \left\{ \lambda_i [g_i(x)]_+ - \frac{\theta_t}{2} \lambda_i^2 \right\} \qquad (7)$$

In this paper, we will use the following assumptions as in [13]: 1. The convex set $S$ is non-empty, closed, bounded, and can be described by $m$ convex functions as $S = \{x : g_i(x) \leq 0, i = 1, 2, \dots, m\}$.

---

**Algorithm 1** Generalized Online Convex Optimization with Long-term Constraint

---
1: **Input:** constraints $g_i(x) \leq 0, i = 1, 2, ..., m$, stepsize $\eta$, time horizon T, and constant $\sigma > 0$.
2: **Initialization:** $x_1$ is in the center of the $\mathcal{B}$ .
3: **for** $t = 1$ **to** $T$ **do**
4:     Input the prediction result $x_t$.
5:     Obtain the convex loss function $f_t(x)$ and the loss value $f_t(x_t)$.
6:     Calculate a subgradient $\partial_x \mathcal{L}_t(x_t, \lambda_t)$, where:

$$\partial_x \mathcal{L}_t(x_t, \lambda_t) = \partial_x f_t(x_t) + \sum_{i=1}^{m} \lambda_t^i \partial_x([g_i(x_t)]_+), \quad \partial_x([g_i(x_t)]_+) = \begin{cases} 0, & g_i(x_t) \leq 0 \\ \nabla_x g_i(x_t), & \text{otherwise} \end{cases}$$

7:     Update $x_t$ and $\lambda_t$ as below:

$$x_{t+1} = \Pi_{\mathcal{B}}(x_t - \eta \partial_x \mathcal{L}_t(x_t, \lambda_t)), \lambda_{t+1} = \frac{[g(x_{t+1})]_+}{\sigma \eta}$$

8: **end for**

---

2. Both the loss functions $f_t(x), \forall t$ and constraint functions $g_i(x), \forall i$ are Lipschitz continuous in the set $\mathcal{B}$. That is, $\|f_t(x) - f_t(y)\| \leq L_f \|x - y\|, \|g_i(x) - g_i(y)\| \leq L_g \|x - y\|, \forall x, y \in \mathcal{B}$ and $\forall t, i$. $G = \max\{L_f, L_g\}$, and

$$F = \max_{t=1,2,...,T} \max_{x,y \in \mathcal{B}} f_t(x) - f_t(y) \leq 2L_f R, \quad D = \max_{i=1,2,...,m} \max_{x \in \mathcal{B}} g_i(x) \leq L_g R$$

## 3  Algorithm

### 3.1  Convex Case:

The main algorithm for this paper is shown in Algorithm 1. For simplicity, we abuse the subgradient notation, denoting a single element of the subgradient by $\partial_x \mathcal{L}_t(x_t, \lambda_t)$. Comparing our algorithm with Eq.(5), we can see that the gradient projection step for $x_{t+1}$ is similar, while the update rule for $\lambda_{t+1}$ is different. Instead of a projected gradient step, we explicitly maximize $\mathcal{L}_{t+1}(x_{t+1}, \lambda)$ over $\lambda$. This explicit projection-free update for $\lambda_{t+1}$ is possible because the constraint clipping guarantees that the maximizer is non-negative. Furthermore, this constraint-violation-dependent update helps to enforce small cumulative and individual constraint violations. Specific bounds on constraint violation are given in Theorem 1 and Lemma 1 below.

Based on the update rule in Algorithm 1, the following theorem gives the upper bounds for both the regret on the loss and the squared-cumulative constraint violation, $\sum_{t=1}^{T} \left([g_i(x_t)]_+\right)^2$ in Problem 6. For space purposes, all proofs are contained in the supplementary material.

**Theorem 1.** *Set $\sigma = \frac{(m+1)G^2}{2(1-\alpha)}$, $\eta = \frac{1}{G\sqrt{(m+1)RT}}$. If we follow the update rule in Algorithm 1 with $\alpha \in (0, 1)$ and $x^*$ being the optimal solution for $\min_{x \in S} \sum_{t=1}^{T} f_t(x)$, we have*

$$\sum_{t=1}^{T} \left(f_t(x_t) - f_t(x^*)\right) \leq O(\sqrt{T}), \quad \sum_{t=1}^{T} \left([g_i(x_t)]_+\right)^2 \leq O(\sqrt{T}), \forall i \in \{1, 2, ..., m\}$$

From Theorem 1, we can see that by setting appropriate stepsize, $\eta$, and constant, $\sigma$, we can obtain the upper bound for the regret of the loss function being less than or equal to $O(\sqrt{T})$, which is also shown in [13] [10]. The main difference of the Theorem 1 is that previous results of [13] [10] all obtain the upper bound for the long-term constraint $\sum_{t=1}^{T} g_i(x_t)$, while here the upper bound for the constraint violation of the form $\sum_{t=1}^{T} \left([g_i(x_t)]_+\right)^2$ is achieved. Also note that the stepsize depends on $T$, which may not be available. In this case, we can use the 'doubling trick' described in the book [2] to transfer our $T$-dependent algorithm into $T$-free one with a worsening factor of $\sqrt{2}/(\sqrt{2} - 1)$.

The proposed algorithm and the resulting bound are useful for two reasons: 1. The square-cumulative constraint implies a bound on the cumulative constraint violation, $\sum_{t=1}^{T} [g_i(x_t)]_+$, while enforcing larger penalties for large violations. 2. The proposed algorithm can also upper bound the constraint violation for each single step $[g_i(x_t)]_+$, which is not bounded in the previous literature.

The next results show how to bound constraint violations at each step.

**Lemma 1.** *If there is only one differentiable constraint function $g(x)$ with Lipschitz continuous gradient parameter $L$, and we run the Algorithm 1 with the parameters in Theorem 1 and large enough $T$, we have*

$$[g(x_t)]_+ \leq O(\tfrac{1}{T^{1/6}}), \quad \forall t \in \{1, 2, ..., T\}, \quad if \quad [g(x_1)]_+ \leq O(\tfrac{1}{T^{1/6}}).$$

Lemma 1 only considers single constraint case. For case of multiple differentiable constraints, we have the following:

**Proposition 1.** *For multiple differentiable constraint functions $g_i(x)$, $i \in \{1, 2, ..., m\}$ with Lipschitz continuous gradient parameters $L_i$, if we use $\bar{g}(x) = \log \left( \sum_{i=1}^{m} \exp g_i(x) \right)$ as the constraint function in Algorithm 1, then for large enough $T$, we have*

$$[g_i(x_t)]_+ \leq O(\tfrac{1}{T^{1/6}}), \quad \forall i, t, \quad if \quad [\bar{g}(x_1)]_+ \leq O(\tfrac{1}{T^{1/6}}).$$

Clearly, both Lemma 1 and Proposition 1 only deal with differentiable functions. For a non-differentiable function $g(x)$, we can first use a differentiable function $\bar{g}(x)$ to approximate the $g(x)$ with $\bar{g}(x) \geq g(x)$, and then apply the previous Lemma 1 and Proposition 1 to upper bound each individual $g_i(x_t)$. Many non-smooth convex functions can be approximated in this way as shown in [16].

## 3.2 Strongly Convex Case:

For $f_t(x)$ to be strongly convex, the Algorithm 1 is still valid. But in order to have lower upper bounds for both objective regret and the clipped long-term constraint $\sum_{t=1}^{T} [g_i(x_t)]_+$ compared with Proposition 3 in next section, we need to use time-varying stepsize as the one used in [9]. Thus, we modify the update rule of $x_t$, $\lambda_t$ to have time-varying stepsize as below:

$$x_{t+1} = \Pi_{\mathcal{B}}(x_t - \eta_t \partial_x \mathcal{L}_t(x_t, \lambda_t)), \quad \lambda_{t+1} = \tfrac{[g(x_{t+1})]_+}{\theta_{t+1}}. \tag{8}$$

If we replace the update rule in Algorithm 1 with Eq.(8), we can obtain the following theorem:

**Theorem 2.** *Assume $f_t(x)$ has strongly convexity parameter $H_1$. If we set $\eta_t = \tfrac{H_1}{t+1}$, $\theta_t = \eta_t(m + 1)G^2$, follow the new update rule in Eq.(8), and $x^*$ being the optimal solution for $\min_{x \in S} \sum_{t=1}^{T} f_t(x)$, for $\forall i \in \{1, 2, ..., m\}$, we have*

$$\sum_{t=1}^{T} \left( f_t(x_t) - f_t(x^*) \right) \leq O(\log(T)), \quad \sum_{t=1}^{T} g_i(x_t) \leq \sum_{t=1}^{T} [g_i(x_t)]_+ \leq O(\sqrt{\log(T)T}).$$

The paper [10] also has a discussion of strongly convex functions, but only provides a bound similar to the convex one. Theorem 2 shows the improved bounds for both objective regret and the constraint violation. On one hand the objective regret is consistent with the standard OCO result in [9], and on the other the constraint violation is further reduced compared with the result in [10].

## 4 Relation with Previous Results

In this section, we extend Theorem 1 to enable direct comparison with the results from [13] [10]. In particular, it is shown how Algorithm 1 recovers the existing regret bounds, while the use of the new augmented Lagrangian (7) in the previous algorithms also provides regret bounds for the clipped constraint case.

The first result puts a bound on the clipped long-term constraint, rather than the sum-of-squares that appears in Theorem 1. This will allow more direct comparisons with the existing results.

**Proposition 2.** *If* $\sigma = \frac{(m+1)G^2}{2(1-\alpha)}$, $\eta = O(\frac{1}{\sqrt{T}})$, $\alpha \in (0,1)$, *and* $x^* = \underset{x \in S}{\arg\min} \sum_{t=1}^{T} f_t(x)$, *then the result of Algorithm 1 satisfies*

$$\sum_{t=1}^{T} \left( f_t(x_t) - f_t(x^*) \right) \leq O(\sqrt{T}), \quad \sum_{t=1}^{T} g_i(x_t) \leq \sum_{t=1}^{T} [g_i(x_t)]_+ \leq O(T^{3/4}), \forall i \in \{1, 2, ..., m\}$$

This result shows that our algorithm generalizes the regret and long-term constraint bounds of [13].

The next result shows that by changing our constant stepsize accordingly, with the Algorithm 1, we can achieve the user-defined trade-off from [10]. Furthermore, we also include the squared version and clipped constraint violations.

**Proposition 3.** *If* $\sigma = \frac{(m+1)G^2}{2(1-\alpha)}$, $\eta = O(\frac{1}{T^\beta})$, $\alpha \in (0,1)$, $\beta \in (0,1)$, *and* $x^* = \underset{x \in S}{\arg\min} \sum_{t=1}^{T} f_t(x)$, *then the result of Algorithm 1 satisfies*

$$\sum_{t=1}^{T} \left( f_t(x_t) - f_t(x^*) \right) \leq O(T^{max\{\beta, 1-\beta\}}),$$
$$\sum_{t=1}^{T} g_i(x_t) \leq \sum_{t=1}^{T} [g_i(x_t)]_+ \leq O(T^{1-\beta/2}), \quad \sum_{t=1}^{T} ([g_i(x_t)]_+)^2 \leq O(T^{1-\beta}), \forall i \in \{1, 2, ..., m\}$$

Proposition 3 provides a systematic way to balance the regret of the objective and the constraint violation. Next, we will show that previous algorithms can use our proposed augmented Lagrangian function to have their own clipped long-term constraint bound.

**Proposition 4.** *If we run Algorithm 1 in [13] with the augmented Lagrangian formula defined in Eq.(7), the result satisfies*

$$\sum_{t=1}^{T} \left( f_t(x_t) - f_t(x^*) \right) \leq O(\sqrt{T}), \quad \sum_{t=1}^{T} g_i(x_t) \leq \sum_{t=1}^{T} [g_i(x_t)]_+ \leq O(T^{3/4}), \forall i \in \{1, 2, ..., m\}.$$

For the update rule proposed in [10], we need to change the $\mathcal{L}_t(x, \lambda)$ to the following one:

$$\mathcal{L}_t(x, \lambda) = f_t(x) + \lambda[g(x)]_+ - \frac{\theta_t}{2}\lambda^2 \tag{9}$$

where $g(x) = \max_{i \in \{1, ..., m\}} g_i(x)$.

**Proposition 5.** *If we use the update rule and the parameter choices in [10] with the augmented Lagrangian in Eq.(9), then* $\forall i \in \{1, ..., m\}$, *we have*

$$\sum_{t=1}^{T} \left( f_t(x_t) - f_t(x^*) \right) \leq O(T^{max\{\beta, 1-\beta\}}), \quad \sum_{t=1}^{T} g_i(x_t) \leq \sum_{t=1}^{T} [g_i(x_t)]_+ \leq O(T^{1-\beta/2}).$$

Propositions 4 and 5 show that clipped long-term constraints can be bounded by combining the algorithms of [13, 10] with our augmented Lagrangian. Although these results are similar in part to our Propositions 2 and 3, they do not imply the results in Theorems 1 and 2 as well as the new single step constraint violation bound in Lemma 1, which are our key contributions. Based on Propositions 4 and 5, it is natural to ask whether we could apply our new augmented Lagrangian formula (7) to the recent work in [19] . Unfortunately, we have not found a way to do so.

Furthermore, since $\left( [g_i(x_t)]_+ \right)^2$ is also convex, we could define $\tilde{g}_i(x_t) = \left( [g_i(x_t)]_+ \right)^2$ and apply the previous algorithms [13] [10] and [19]. This will result in the upper bounds of $O(T^{3/4})$ [13] and $O(T^{1-\beta/2})$ [10], which are worse than our upper bounds of $O(T^{1/2})$ (Theorem 1) and $O(T^{1-\beta})$ ( Proposition 3). Note that the algorithm in [19] cannot be applied since the clipped constraints do not satisfy the required Slater condition.

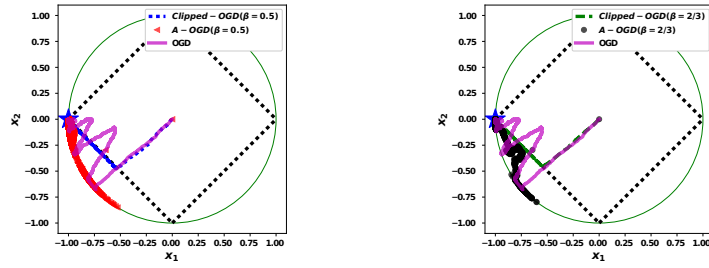

Figure 1: Toy Example Results: Trajectories generated by different algorithms. Note how trajectories generated by Clipped-OGD follow the desired constraints tightly. In contrast, OGD oscillates around the true constraints, and A-OGD closely follows the boundary of the outer ball.

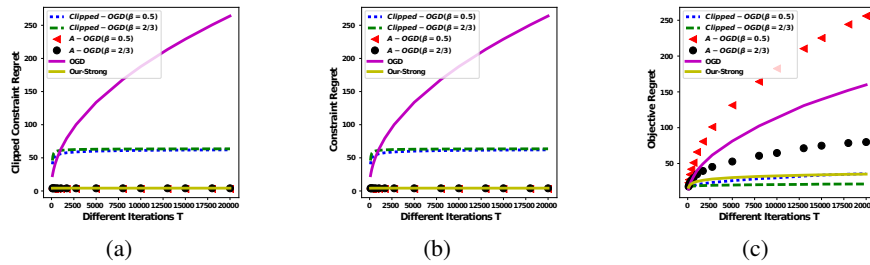

(a)  (b)  (c)

Figure 2: **Doubly-Stochastic Matrices.** Fig.2(a): Clipped Long-term Constraint Violation. Fig.2(b): Long-term Constraint Violation. Fig.2(c): Cumulative Regret of the Loss function

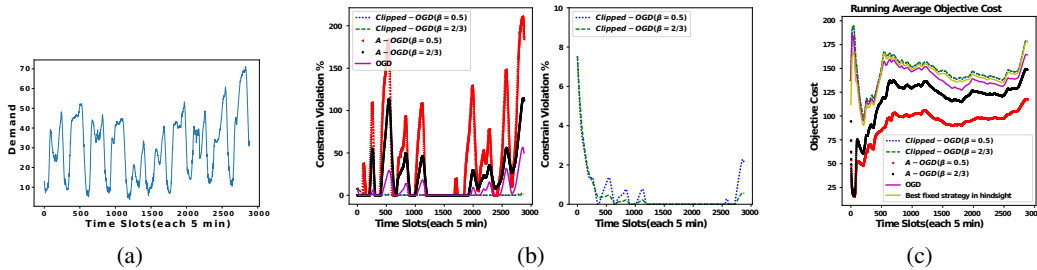

(a)  (b)  (c)

Figure 3: **Economic Dispatch.** Fig.3(a): Power Demand Trajectory. Fig.3(b): Constraint Violation for each time step. All of the previous algorithms incurred substantial constraint violations. The figure on the right shows the violations of our algorithm, which are significantly smaller. Fig.3(c): Running Average of the Objective Loss

# 5  Experiments

In this section, we test the performance of the algorithms including OGD [13], A-OGD [10], Clipped-OGD (this paper), and our proposed algorithm strongly convex case (Our-strong). Throughout the experiments, our algorithm has the following fixed parameters: $\alpha = 0.5$, $\sigma = \frac{(m+1)G^2}{2(1-\alpha)}$, $\eta = \frac{1}{T^\beta G\sqrt{R(m+1)}}$. In order to better show the result of the constraint violation trajectories, we aggregate all the constraints as a single one by using $g(x_t) = \max_{i \in \{1,\dots,m\}} g_i(x_t)$ as done in [13].

## 5.1  A Toy Experiment

For illustration purposes, we solve the following 2-D toy experiment with $x = [x_1, x_2]^T$:

$$\min \sum_{t=1}^{T} c_t^T x, \quad s.t. \, |x_1| + |x_2| - 1 \leq 0. \tag{10}$$

where the constraint is the $\ell_1$-norm constraint. The vector $c_t$ is generated from a uniform random vector over $[0, 1.2] \times [0, 1]$ which is rescaled to have norm 1. This leads to slightly average cost on the on the first coordinate. The offline solutions for different $T$ are obtained by CVXPY [5].

All algorithms are run up to $T = 20000$ and are averaged over 10 random sequences of $\{c_t\}_{t=1}^{T}$. Since the main goal here is to compare the variables' trajectories generated by different algorithms, the results for different $T$ are in the supplementary material for space purposes. Fig.1 shows these trajectories for one realization with $T = 8000$. The blue star is the optimal point's position.

From Fig.1 we can see that the trajectories generated by Clipped-OGD follows the boundary very tightly until reaching the optimal point. This can be explained by the Lemma 1 which shows that the constraint violation for single step is also upper bounded. For the OGD, the trajectory oscillates widely around the boundary of the true constraint. For the A-OGD, its trajectory in Fig.1 violates the constraint most of the time, and this violation actually contributes to the lower objective regret shown in the supplementary material.

## 5.2  Doubly-Stochastic Matrices

We also test the algorithms for approximation by doubly-stochastic matrices, as in [10]:

$$\min \sum_{t=1}^{T} \tfrac{1}{2} \|Y_t - X\|_F^2 \quad s.t. \quad X\mathbf{1} = \mathbf{1}, \quad X^T\mathbf{1} = \mathbf{1}, \quad X_{ij} \geq 0. \tag{11}$$

where $X \in \mathbb{R}^{d \times d}$ is the matrix variable, $\mathbf{1}$ is the vector whose elements are all 1, and matrix $Y_t$ is the permutation matrix which is randomly generated.

After changing the equality constraints into inequality ones (e.g.,$X\mathbf{1} = \mathbf{1}$ into $X\mathbf{1} \geq \mathbf{1}$ and $X\mathbf{1} \leq \mathbf{1}$), we run the algorithms with different T up to $T = 20000$ for 10 different random sequences of $\{Y_t\}_{t=1}^{T}$. Since the objective function $f_t(x)$ is strongly convex with parameter $H_1 = 1$, we also include our designed strongly convex algorithm as another comparison. The offline optimal solutions are obtained by CVXPY [5].

The mean results for both constraint violation and objective regret are shown in Fig.2. From the result we can see that, for our designed strongly convex algorithm Our-Strong, its result is around the best ones in not only the clipped constraint violation, but the objective regret. For our most-balanced convex case algorithm Clipped-OGD with $\beta = 0.5$, although its clipped constraint violation is relatively bigger than A-OGD, it also becomes quite flat quickly, which means the algorithm quickly converges to a feasible solution.

## 5.3  Economic Dispatch in Power Systems

This example is adapted from [12] and [18], which considers the problem of power dispatch. That is, at each time step $t$, we try to minimize the power generation cost $c_i(x_{t,i})$ for each generator $i$ while maintaining the power balance $\sum_{i=1}^{n} x_{t,i} = d_t$, where $d_t$ is the power demand at time $t$. Also, each power generator produces an emission level $E_i(x_{t,i})$. To bound the emissions, we impose the

constraint $\sum\limits_{i=1}^{n} E_i(x_{t,i}) \leq E_{max}$. In addition to requiring this constraint to be satisfied on average, we also require bounded constraint violations at each timestep. The problem is formally stated as:

$$\min \sum_{t=1}^{T} \Big( \sum_{i=1}^{n} c_i(x_{t,i}) + \xi(\sum_{i=1}^{n} x_{t,i} - d_t)^2 \Big), \quad s.t. \quad \sum_{i=1}^{n} E_i(t,i) \leq E_{max}, \quad 0 \leq x_{t,i} \leq x_{i,max}.$$
(12)

where the second constraint is from the fact that each generator has the power generation limit.

In this example, we use three generators. We define the cost and emission functions according to [18] and [12] as $c_i(x_{t,i}) = 0.5a_i x_{t,i}^2 + b_i x_{t,i}$, and $E_i = d_i x_{t,i}^2 + e_i x_{t,i}$, respectively. The parameters are: $a_1 = 0.2, a_2 = 0.12, a_3 = 0.14, b_1 = 1.5, b_2 = 1, b_3 = 0.6, d_1 = 0.26, d_2 = 0.38, d_3 = 0.37$, $E_{max} = 100, \xi = 0.5$, and $x_{1,max} = 20, x_{2,max} = 15, x_{3,max} = 18$. The demand $d_t$ is adapted from real-world 5-minute interval demand data between 04/24/2018 and 05/03/2018 [1], which is shown in Fig.3(a). The offline optimal solution or best fixed strategy in hindsight is obtained by an implementation of SAGA [4]. The constraint violation for each time step is shown in Fig.3(b), and the running average objective cost is shown in Fig.3(c). From these results we can see that our algorithm has very small constraint violation for each time step, which is desired by the requirement. Furthermore, our objective costs are very close to the best fixed strategy.

## 6  Conclusion

In this paper, we propose two algorithms for OCO with both convex and strongly convex objective functions. By applying different update strategies that utilize a modified augmented Lagrangian function, they can solve OCO with a squared/clipped long-term constraints requirement. The algorithm for general convex case provides the useful bounds for both the long-term constraint violation and the constraint violation at each timestep. Furthermore, the bounds for the strongly convex case is an improvement compared with the previous efforts in the literature. Experiments show that our algorithms can follow the constraint boundary tightly and have relatively smaller clipped long-term constraint violation with reasonably low objective regret. It would be useful if future work could explore the noisy versions of the constraints and obtain the similar upper bounds.

## Acknowledgments

Thanks to Tianyi Chen for valuable discussions about algorithm's properties.

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
