[Supplementary Material · neurips_2018-supplementary.pdf]



Figure 4: Toy Example Results: Fig.4(a): Clipped Long-term Constraint Violation. Fig.4(b): Long-term Constraint Violation. Fig.4(c): Cumulative Regret of the Loss function

## Supplemental Materials

The supplemental material contains proofs of the main results of the paper along with supporting results.

## A  Toy Example Results

The results including different $T$ up to 20000 are shown in Fig.4, whose results are averaged over 10 random sequences of $\{c_t\}_{t=1}^T$. Since the standard deviations are small, we only plot the mean results.

From Fig.1 we can see that the trajectories generated by $Clipped - OGD$ follows the boundary very tightly until reaching the optimal point. which is also reflected by the Fig.4(a) of the clipped long-term constraint violation. For the $OGD$, its trajectory oscillates a lot around the boundary of the actual constraint. And if we examine the clipped and non-clipped constraint violation in Fig.4, we find that although the clipped constraint violation is very high, its non-clipped one is very small. This verifies the statement we make in the beginning that the big constraint violation at one time step is canceled out by the strictly feasible constraint at the other time step. For the $A - OGD$, its trajectory in Fig.1 violates the constraint most of the time, and this violation actually contributes to the lower objective regret shown in Fig.4.

## B  Proof of Theorem 1

Before proving Theorem 1, we need the following preliminary result.

**Lemma 2.** *For the sequence of $x_t$, $\lambda_t$ obtained from Algorithm 1 and $\forall x \in \mathcal{B}$, we can prove the following inequality:*

$$
\begin{aligned}
\sum_{t=1}^T [\mathcal{L}_t(x_t, \lambda_t) - \mathcal{L}_t(x, \lambda_t)] \ &\leq \tfrac{R^2}{2\eta} + \tfrac{\eta T}{2}(m+1)G^2 \\
&+ \tfrac{\eta}{2}(m+1)G^2 \sum_{t=1}^T \|\lambda_t\|^2
\end{aligned}
\tag{13}
$$

*Proof.* First, $\mathcal{L}_t(x, \lambda)$ is convex in $x$. Then for any $x \in \mathcal{B}$, we have the following inequality:

$$
\mathcal{L}_t(x_t, \lambda_t) - \mathcal{L}_t(x, \lambda_t) \leq (x_t - x)^T \partial_x \mathcal{L}_t(x_t, \lambda_t)
\tag{14}
$$

Using the non-expansive property of the projection operator and the update rule for $x_{t+1}$ in Algorithm 1, we have

$$
\begin{aligned}
\|x - x_{t+1}\|^2 \ &\leq \|x - (x_t - \eta \partial_x \mathcal{L}_t(x_t, \lambda_t))\|^2 \\
&= \|x - x_t\|^2 - 2\eta(x_t - x)^T \partial_x \mathcal{L}_t(x_t, \lambda_t) \\
&\quad + \eta^2 \|\partial_x \mathcal{L}_t(x_t, \lambda_t)\|^2
\end{aligned}
\tag{15}
$$

Then we have

$$\mathcal{L}_t(x_t, \lambda_t) - \mathcal{L}_t(x, \lambda_t) \le \frac{1}{2\eta}\left(\|x - x_t\|^2 - \|x - x_{t+1}\|^2\right) + \frac{\eta}{2}\|\partial_x \mathcal{L}_t(x_t, \lambda_t)\|^2 \tag{16}$$

Furthermore, for $\|\partial_x \mathcal{L}_t(x_t, \lambda_t)\|^2$, we have

$$\|\partial_x \mathcal{L}_t(x_t, \lambda_t)\|^2 = \left\|\partial_x f_t(x_t) + \sum_{i=1}^{m} \lambda_t^i \partial_x([g_i(x_t)]_+)\right\|^2 \le (m+1)G^2(1 + \|\lambda_t\|^2) \tag{17}$$

where the last inequality is from the inequality that $(y_1 + y_2 + ... + y_n)^2 \le n(y_1^2 + y_2^2 + ... + y_n^2)$, and both $\|\partial_x f_t(x_t)\|$ and $\|\partial_x([g_i(x_t)]_+)\|$ are less than or equal to $G$ by the definition.

Then we have

$$\mathcal{L}_t(x_t, \lambda_t) - \mathcal{L}_t(x, \lambda_t) \le \frac{1}{2\eta}\left(\|x - x_t\|^2 - \|x - x_{t+1}\|^2\right) + \frac{\eta}{2}(m+1)G^2(1 + \|\lambda_t\|^2) \tag{18}$$

Since $x_1$ is in the center of $\mathcal{B}$, we can assume $x_1 = 0$ without loss of generality. If we sum the $\mathcal{L}_t(x_t, \lambda_t) - \mathcal{L}_t(x, \lambda_t)$ from 1 to $T$, we have

$$\begin{aligned}\sum_{t=1}^{T}[\mathcal{L}_t(x_t, \lambda_t) - \mathcal{L}_t(x, \lambda_t)] &\le \frac{1}{2\eta}\left(\|x - x_1\|^2 - \|x - x_{T+1}\|^2\right) \\ &\quad + \frac{\eta T}{2}(m+1)G^2 \\ &\quad + \frac{\eta}{2}(m+1)G^2 \sum_{t=1}^{T}\|\lambda_t\|^2 \\ &\le \frac{R^2}{2\eta} + \frac{\eta T}{2}(m+1)G^2 \\ &\quad + \frac{\eta}{2}(m+1)G^2 \sum_{t=1}^{T}\|\lambda_t\|^2\end{aligned} \tag{19}$$

where the last inequality follows from the fact that $x_1 = 0$ and $\|x\|^2 \le R^2$. $\qquad\square$

Now we are ready to prove the main theorem.

*Proof of Theorem 1.* From Lemma 2, we have

$$\begin{aligned}\sum_{t=1}^{T}[\mathcal{L}_t(x_t, \lambda_t) - \mathcal{L}_t(x, \lambda_t)] &\le \frac{R^2}{2\eta} + \frac{\eta T}{2}(m+1)G^2 \\ &\quad + \frac{\eta}{2}(m+1)G^2 \sum_{t=1}^{T}\|\lambda_t\|^2\end{aligned} \tag{20}$$

If we expand the terms in the left-hand side and move the last term in right-hand side to the left, we have

$$\begin{aligned}&\sum_{t=1}^{T}\left(f_t(x_t) - f_t(x)\right) + \sum_{t=1}^{T}\sum_{i=1}^{m}\left(\lambda_t^i[g_i(x_t)]_+ - \lambda_t^i[g_i(x)]_+\right) \\ &- \frac{\eta}{2}(m+1)G^2 \sum_{t=1}^{T}\|\lambda_t\|^2 \le \frac{R^2}{2\eta} + \frac{\eta T}{2}(m+1)G^2\end{aligned} \tag{21}$$

If we set $x = x^*$ to have $[g_i(x^*)]_+ = 0$ and plug in the expression $\lambda_t = \frac{[g(x_t)]_+}{\sigma\eta}$, we have

$$\begin{aligned}\sum_{t=1}^{T}\left(f_t(x_t) - f_t(x^*)\right) &+ \sum_{i=1}^{m}\sum_{t=1}^{T}\frac{([g_i(x_t)]_+)^2}{\sigma\eta}\left(1 - \frac{(m+1)G^2}{2\sigma}\right) \\ &\le \frac{R^2}{2\eta} + \frac{\eta T}{2}(m+1)G^2\end{aligned} \tag{22}$$

If we plug in the expression for $\sigma$ and $\eta$, we have

$$\begin{aligned}\sum_{t=1}^{T}\left(f_t(x_t) - f_t(x^*)\right) &+ \sum_{i=1}^{m}\sum_{t=1}^{T}\frac{([g_i(x_t)]_+)^2}{\sigma\eta}\alpha \\ &\le O(\sqrt{T})\end{aligned} \tag{23}$$

Because $\frac{([g_i(x_t)]_+)^2}{\sigma\eta}\alpha \geq 0$, we have

$$\sum_{t=1}^{T}\Big(f_t(x_t) - f_t(x^*)\Big) \leq O(\sqrt{T}) \tag{24}$$

Furthermore, we have $\sum_{t=1}^{T}\Big(f_t(x_t) - f_t(x^*)\Big) \geq -FT$ according to the assumption. Then we have

$$\sum_{i=1}^{m}\sum_{t=1}^{T}\Big([g_i(x_t)]_+\Big)^2 \leq \frac{\sigma\eta}{\alpha}(O(\sqrt{T}) + FT)$$
$$= \frac{\sigma}{\alpha}(O(\sqrt{T}) + FT)O(\frac{1}{\sqrt{T}}) = O(\sqrt{T}) \tag{25}$$

Because $\Big([g_i(x_t)]_+\Big)^2 \geq 0$, we have

$$\sum_{t=1}^{T}\Big([g_i(x_t)]_+\Big)^2 \leq O(\sqrt{T}), \forall i \in \{1,2,...,m\} \tag{26}$$

$\square$

## C  Proof of Lemma 1

*Proof.* Recall that the update for $x_{t+1}$ is

$$x_{t+1} = \Pi_\mathcal{B}\Big(x_t - \eta\partial_x f_t(x_t) - \frac{[g(x_t)]_+}{\sigma}\partial_x([g(x_t)]_+)\Big) \tag{27}$$

Let $y_t = x_t - \eta\partial_x f_t(x_t) - \frac{[g(x_t)]_+}{\sigma}\partial_x([g(x_t)]_+)$.

We first need to show that $g(x_{t+1}) \leq g(y_t)$. Without loss of generality, let us assume that $y_t$ is not in the set $\mathcal{B}$. From convexity we have $g(y_t) \geq g(x_{t+1}) + \nabla_x g(x_{t+1})^T(y_t - x_{t+1})$. From non-expansiveness of the projection operator, we have that $(y_t-x_{t+1})^T(x-x_{t+1}) \leq 0$ for $x \in \mathcal{B}$. Let $x = x_{t+1}-\epsilon_0\nabla_x g(x_{t+1})$ with $\epsilon_0$ small enough to make $x \in \mathcal{B}$. We have $-\epsilon_0(y_t-x_{t+1})^T\nabla_x g(x_{t+1}) \leq 0$. Then we have $g(x_{t+1}) \leq g(y_t)$.

As a result, if $g(y_t)$ is upper bounded, then so is $g(x_{t+1})$, where $x_{t+1} = \Pi_\mathcal{B}(y_t)$. If $T$ is large enough, $\eta\,\|\partial_x f_t(x_t)\|$ would be very small. Thus, we can use 0-order Taylor expansion for differentiable $g(x)$ as below:

$$g(y_t) = g\Big(x_t - \eta\partial_x f_t(x_t) - \frac{[g(x_t)]_+}{\sigma}\partial_x([g(x_t)]_+)\Big) \tag{28}$$
$$\leq g\Big(x_t - \frac{[g(x_t)]_+}{\sigma}\partial_x([g(x_t)]_+)\Big) + C\eta$$

where $C$ is a constant determined by the Taylor expansion remainder, as well as the bound $\|\partial_x[g(x_t)]_+\|\|\partial_X f(x_t)\| \leq G^2$.

Set $\epsilon = (2C\sigma R^2\eta)^{1/3} = O(\frac{1}{T^{1/6}})$. We will show that if $g(x_t) < \epsilon$, then $g(x_{t+1}) \leq \epsilon+O(1/\sqrt{T}) = O(\frac{1}{T^{1/6}})$. We will also show that if $g(x_t) \geq \epsilon$, then $g(x_{t+1}) \leq g(x_t)$. It follows then by induction that if $g(x_1) < \epsilon$, then $g(x_t) = O(\frac{1}{T^{1/6}})$ for all $t$. We prove these inequalities in three cases. Since $g(x_{t+1}) \leq g(y_t)$, it suffices to bound $g(y_t)$.

**Case 1:** $g(x_t) \leq 0$. In this case, the inequality for $g(y_t)$, (28), becomes

$$g(y_t) \leq g(x_t) + C\eta \leq C\eta = O(\frac{1}{\sqrt{T}}) \tag{29}$$

**Case 2:** $0 < g(x_t) < \epsilon$. Since $[g(x_t)]_+ = g(x_t)$, the bound on $g(y_t)$ becomes

$$g(y_t) \leq g\Big(x_t - \frac{g(x_t)}{\sigma}\nabla_x g(x_t)\Big) + C\eta \tag{30}$$

We will bound the right using standard methods from gradient descent proofs. Since $g$ is convex and $\nabla_x g(x)$ has Lipschitz constant, $L$, we have the inequality:

$$g(y) \leq g(x) + \nabla_x g(x)^T (y - x) + \frac{L}{2} \|y - x\|^2 \tag{31}$$

for all $x$ and $y$ [17].

Recall that $\epsilon = O(\frac{1}{T^{1/6}})$. Assume that $T$ is sufficiently large so that $\frac{Lg(x_t)}{2\sigma} < \frac{L\epsilon}{2\sigma} < 1$. Applying (31) with $x = x_t$ and $y = x_t - \frac{g(x_t)}{\sigma}\nabla_x g(x_t)$ gives

$$g(y_t) \leq g\left(x_t - \frac{[g(x_t)]_+}{\sigma}\partial_x(g(x_t))\right) + C\eta \tag{32}$$

$$\leq g(x_t) - \frac{g(x_t)}{\sigma}(1 - \frac{Lg(x_t)}{2\sigma})\|\nabla_x g(x_t)\|^2 + C\eta \tag{33}$$

$$\leq g(x_t) + C\eta = O(\frac{1}{T^{1/6}}). \tag{34}$$

where the third bound follows since $1 - \frac{Lg(x_t)}{2\sigma} > 0$.

**Case 3:** $g(x_t) \geq \epsilon$. A case can arise such that $g(x_{t-1}) < \epsilon$ but an additive term of order $O(\frac{1}{T^{1/2}})$ leads to $\epsilon \leq g(x_t) \leq \epsilon + C\eta = O(\frac{1}{T^{1/6}})$. We will now show that no further increases are possible by bounding the final two terms of (33) as

$$-\frac{g(x_t)}{\sigma}(1 - \frac{Lg(x_t)}{2\sigma})\|\nabla_x g(x_t)\|^2 + C\eta \leq 0 \iff C\eta \leq \frac{g(x_t)}{\sigma}(1 - \frac{Lg(x_t)}{2\sigma})\|\nabla_x g(x_t)\|^2. \tag{35}$$

Now, we lower-bound the terms on the right of (35). Since $\epsilon + C\eta = O(\frac{1}{T^{1/6}})$, we have that for sufficiently large $T$, $1 - \frac{Lg(x_t)}{2\sigma} \geq 1 - \frac{L(\epsilon + C\eta)}{2\sigma} \geq \frac{1}{2}$. Further note that by convexity, $g(0) \geq g(x_t) - \nabla_x g(x_t)^T x_t$. Since we assume that $0$ is feasible, we have that

$$\epsilon \leq g(x_t) \leq \nabla_x g(x_t)^T x_t \leq \|\nabla_x g(x_t)\|\|x_t\| \leq \|\nabla_x g(x_t)\|R.$$

The final inequality follows since $x_t \in \mathcal{B}$. Thus, we have the following bound for the right of (35):

$$\frac{g(x_t)}{\sigma}(1 - \frac{Lg(x_t)}{2\sigma})\|\nabla_x g(x_t)\|^2 \geq \frac{\epsilon^3}{2\sigma R^2} = C\eta.$$

The final equality follows by the definition of $\epsilon$. $\qquad\square$

## D  Proof of Theorem 2

*Proof.* For the strongly convex case of $f_t(x)$ with strong convexity parameter equal to $H_1$, we can also conclude that the modified augmented Lagrangian function in Eq.(8) is also strongly convex w.r.t. $x$ with the strong convexity parameter $H \geq H_1$. Then we have

$$\begin{aligned} \mathcal{L}_t(x^*, \lambda_t) - \mathcal{L}_t(x_t, \lambda_t) \quad &\geq \partial_x \mathcal{L}_t(x_t)^T(x^* - x_t) \\ &+ \frac{H_1}{2}\|x^* - x_t\|^2 \end{aligned} \tag{36}$$

From concavity of $\mathcal{L}$ in terms of $\lambda$, we can have

$$\mathcal{L}_t(x_t, \lambda) - \mathcal{L}_t(x_t, \lambda_t) \leq (\lambda - \lambda_t)^T \nabla_\lambda \mathcal{L}_t(x_t, \lambda_t) \tag{37}$$

Since $\lambda_t$ maximizes the augmented Lagrangian, we can see that the right hand side is $0$.

From Eq.(15), we have

$$\begin{aligned} \partial_x \mathcal{L}_t(x_t)^T(x_t - x^*) \leq \quad &\frac{1}{2\eta_t}\left(\|x^* - x_t\|^2 - \|x^* - x_{t+1}\|^2\right) \\ &+ \frac{\eta_t}{2}(m+1)G^2(1 + \|\lambda_t\|^2) \end{aligned} \tag{38}$$

Multiply Eq.(36) by $-1$ and add Eq.(37) together with Eq.(38) plugging in:

$$\begin{aligned} \mathcal{L}_t(x_t, \lambda) - \mathcal{L}_t(x^*, \lambda_t) \leq \quad &\frac{1}{2\eta_t}\left(\|x^* - x_t\|^2 - \|x^* - x_{t+1}\|^2\right) \\ &+ \frac{\eta_t}{2}(m+1)G^2(1 + \|\lambda_t\|^2) - \frac{H_1}{2}\|x^* - x_t\|^2 \end{aligned} \tag{39}$$

Let $b_t = \|x^* - x_t\|^2$, and plug in the expression for $\mathcal{L}_t$, we can get:

$$
\begin{aligned}
&f_t(x_t) - f_t(x^*) + \lambda^T [g(x_t)]_+ - \tfrac{\theta_t}{2}\|\lambda\|^2 \le \tfrac{1}{2\eta_t}(b_t - b_{t+1}) \\
&\quad - \tfrac{H_1}{2} b_t + \tfrac{(m+1)G^2}{2}\eta_t + \tfrac{(m+1)G^2}{2}\|\lambda_t\|^2 \left(\eta_t - \tfrac{\theta_t}{(m+1)G^2}\right)
\end{aligned}
\tag{40}
$$

Plug in the expressions $\eta_t = \frac{H_1}{t+1}$, $\theta_t = (m+1)G^2 \eta_t$, and sum over $t = 1$ to $T$:

$$
\begin{aligned}
&\sum_{t=1}^{T}\Big(f_t(x_t) - f_t(x^*)\Big) + \lambda^T\Big(\sum_{t=1}^{T}[g(x_t)]_+\Big) - \tfrac{\|\lambda\|^2}{2}\sum_{t=1}^{T}\theta_t \\
&\le \underbrace{\tfrac{1}{2}\sum_{t=1}^{T}\left(\tfrac{b_t - b_{t+1}}{\eta_t} - \tfrac{H_1}{2}b_t\right)}_{A} + \underbrace{\tfrac{(m+1)G^2}{2}\sum_{t=1}^{T}\eta_t}_{B}
\end{aligned}
\tag{41}
$$

For the expression of $A$, we have:

$$
\begin{aligned}
A &= \tfrac{1}{2}\left[\tfrac{b_1}{\eta_1} + \sum_{t=2}^{T} b_t\left(\tfrac{1}{\eta_t} - \tfrac{1}{\eta_{t-1}} - H_1\right) - \tfrac{b_{T+1}}{\eta_T} - H_1 b_1\right] \\
&\le \tfrac{b_1}{H_1}
\end{aligned}
\tag{42}
$$

For the expression of $B$, with the expression of $\eta_t$ and the inequality relation between sum and integral, we have:

$$
B \le \frac{(m+1)G^2 H_1}{2}\log(T)
\tag{43}
$$

Thus, we have:

$$
\begin{aligned}
&\sum_{t=1}^{T}\Big(f_t(x_t) - f_t(x^*)\Big) + \lambda^T\Big(\sum_{t=1}^{T}[g(x_t)]_+\Big) - \tfrac{\|\lambda\|^2}{2}\sum_{t=1}^{T}\theta_t \\
&\le O(\log(T))
\end{aligned}
\tag{44}
$$

If we set $\lambda = \dfrac{\sum_{t=1}^{T}[g(x_t)]_+}{\sum_{t=1}^{T}\theta_t}$, and due to non-negativity of $\dfrac{\left\|\sum_{t=1}^{T}[g(x_t)]_+\right\|^2}{2\sum_{t=1}^{T}\theta_t}$, we can have

$$
\sum_{t=1}^{T}\Big(f_t(x_t) - f_t(x^*)\Big) \le O(\log(T))
\tag{45}
$$

Furthermore, we have $\sum_{t=1}^{T}\Big(f_t(x_t) - f_t(x^*)\Big) \ge -FT$ according to the assumption. Then we have

$$
\frac{\left\|\sum_{t=1}^{T}[g(x_t)]_+\right\|^2}{2\sum_{t=1}^{T}\theta_t} \le O(\log(T)) + FT
\tag{46}
$$

Because $\sum_{t=1}^{T}\theta_t \le (m+1)G^2 H_1 \log(T)$, we have:

$$
\left\|\sum_{t=1}^{T}[g(x_t)]_+\right\| \le O(\sqrt{\log(T)T})
\tag{47}
$$

$\square$

# E   Proof of the Propositions

Now we give the proofs for all the remaining Propositions.

*Proof of the Proposition 1.* From the construction of $\bar{g}(x)$, we have the $\bar{g}(x) \geq \max_i g_i(x)$. Thus, if we can upper bound the $\bar{g}(x)$, $g_i(x)$ will automatically be upper bounded. In order to use Lemma 1, we need to make sure the following conditions are satisfied:

- $\bar{g}(x)$ is convex and differentiable.

- $\|\nabla_x \bar{g}(x)\|$ is upper bounded.

- $\|\nabla_x'' \bar{g}(x)\|_2$ is upper bounded.

The first condition is satisfied due to the formula of $\bar{g}(x)$. To examine the second one, we have

$$\nabla_x \bar{g}(x) = \frac{1}{\sum\limits_{i=1}^{m} \exp g_i(x)} \left[ \sum_{i=1}^{m} \exp g_i(x) \nabla_x g_i(x) \right] \tag{48}$$

$$
\begin{aligned}
\|\nabla_x \bar{g}(x)\|^2 &= \frac{1}{\left( \sum\limits_{i=1}^{m} \exp g_i(x) \right)^2} \left\| \sum_{i=1}^{m} \exp g_i(x) \nabla_x g_i(x) \right\|^2 \\
&\leq \frac{m \sum\limits_{i=1}^{m} (\exp g_i(x))^2 \|\nabla_x g_i(x)\|^2}{\left( \sum\limits_{i=1}^{m} \exp g_i(x) \right)^2} \\
&\leq m G^2
\end{aligned}
\tag{49}
$$

Thus, $\|\nabla_x \bar{g}(x)\| \leq \sqrt{m} G$ and the second condition is satisfied.

For $\|\nabla_x'' \bar{g}(x)\|_2$, we have

$$
\begin{aligned}
\nabla_x'' \bar{g}(x) = \quad & \underbrace{\frac{1}{\sum\limits_{i=1}^{m} \exp g_i(x)} \left[ \sum_{i=1}^{m} \exp g_i(x) \nabla_x'' g_i(x) + \exp g_i(x) \nabla_x g_i(x) \nabla_x g_i(x)^T \right]}_{A} \\
& - \underbrace{\frac{1}{\sum\limits_{i=1}^{m} \exp g_i(x)} \left( \sum_{i=1}^{m} \exp g_i(x) \nabla_x g_i(x) \right) \left( \sum_{i=1}^{m} \exp g_i(x) \nabla_x g_i(x)^T \right)}_{B}
\end{aligned}
\tag{50}
$$

To upper bound $\|\nabla_x'' \bar{g}(x)\|_2$, which is

$$\max_{u^T u = 1} u^T \nabla_x'' \bar{g}(x) u = \max_{u^T u = 1} u^T A u - u^T B u \leq \max_{u^T u = 1} u^T A u \tag{51}$$

where the inequality is due to the fact that $B \succeq 0$.

Thus, we have $\|\nabla_x'' \bar{g}(x)\|_2 \leq \|A\|_2$. For the $\|A\|_2$, we have

$$
\begin{aligned}
\|A\|_2 = \max_{u^T u = 1} u^T A u &\leq \frac{1}{\sum\limits_{i=1}^{m} \exp g_i(x)} \left( \sum_{i=1}^{m} \max_{u^T u = 1} \exp g_i(x) u^T \nabla_x'' g_i(x) u \right) \\
&+ \frac{1}{\sum\limits_{i=1}^{m} \exp g_i(x)} \left( \sum_{i=1}^{m} \max_{u^T u = 1} \exp g_i(x) \left\| \nabla_x g_i(x)^T u \right\|^2 \right) \\
&\leq \frac{1}{\sum\limits_{i=1}^{m} \exp g_i(x)} \left( \sum_{i=1}^{m} \exp g_i(x)(L_i + \|\nabla_x g_i(x)\|^2) \right) \\
&\leq \frac{1}{\sum\limits_{i=1}^{m} \exp g_i(x)} \left( \sum_{i=1}^{m} \exp g_i(x) \right) (\bar{L} + G^2) = \bar{L} + G^2
\end{aligned}
\tag{52}
$$

where the first inequality comes from the optimality definition, the second inequality comes from the upper bound for each $\|\nabla''_x g_i(x)\|_2$ and the Cauchy - Schwartz inequality, and the last inequality comes from the fact that $\bar{L} = \max L_i$ and $\|\nabla_x g_i(x)\|$ is upper bounded by $G$. Thus, the last condition is also satisfied. $\qquad\square$

*Proof of the Proposition 2.* From Theorem 1, we know that $\sum_{t=1}^{T} \left([g_i(x_t)]_+\right)^2 \leq O(\sqrt{T})$. By using the inequality $(y_1 + y_2 + ... + y_n)^2 \leq n(y_1^2 + y_2^2 + ... + y_n^2)$, setting $y_i$ being equal to $[g_i(x_t)]_+$, and $n = T$, we have $\left(\sum_{t=1}^{T} [g_i(x_t)]_+\right)^2 \leq T \sum_{t=1}^{T} \left([g_i(x_t)]_+\right)^2 \leq O(T^{3/2})$. Then we obtain that

$\sum_{t=1}^{T} [g_i(x_t)]_+ \leq O(T^{3/4})$. Because $g_i(x_t) \leq [g_i(x_t)]_+$, we also have $g_i(x_t) \leq O(T^{3/4})$. $\qquad\square$

*Proof of the Proposition 3.* Since we only change the stepsize for Algorithm 1, the previous result in Lemma 2 and part of the proof up to Eq.(22) in Theorem 1 can be used without any changes.

First, let us rewrite the Eq.(22):

$$\begin{aligned}
\sum_{t=1}^{T} \left(f_t(x_t) - f_t(x^*)\right) \quad &+ \sum_{i=1}^{m} \sum_{t=1}^{T} \frac{([g_i(x_t)]_+)^2}{\sigma\eta}\left(1 - \frac{(m+1)G^2}{2\sigma}\right) \\
&\leq \frac{R^2}{2\eta} + \frac{\eta T}{2}(m+1)G^2
\end{aligned} \tag{53}$$

By plugging in the definition of $\alpha$, $\eta$, and that $\frac{([g_i(x_t)]_+)^2}{\sigma\eta}\alpha \geq 0$, we have

$$\begin{aligned}
\sum_{t=1}^{T} \left(f_t(x_t) - f_t(x^*)\right) \quad &\leq \frac{R^2}{2}T^\beta + \frac{(m+1)G^2}{2}T^{1-\beta} \\
&= O(T^{max\{\beta, 1-\beta\}})
\end{aligned} \tag{54}$$

As argued in the proof of Theorem 1, we have the following inequality with the help of $\sum_{t=1}^{T} \left(f_t(x_t) - f_t(x^*)\right) \geq -FT$:

$$\begin{aligned}
\sum_{i=1}^{m} \sum_{t=1}^{T} \frac{([g_i(x_t)]_+)^2}{\sigma\eta}\alpha &\leq \frac{R^2}{2}T^\beta + \frac{(m+1)G^2}{2}T^{1-\beta} + FT \\
\sum_{t=1}^{T} ([g_i(x_t)]_+)^2 &\leq \frac{\sigma}{\alpha}\left(\frac{R^2}{2} + \frac{(m+1)G^2}{2}T^{1-2\beta} + FT^{1-\beta}\right)
\end{aligned} \tag{55}$$

Then we have

$$\begin{aligned}
\sum_{t=1}^{T} [g_i(x_t)]_+ &\leq \sqrt{T \sum_{t=1}^{T} \left([g_i(x_t)]_+\right)^2} \\
&\leq \sqrt{\frac{T\sigma}{\alpha}\left(\frac{R^2}{2} + \frac{(m+1)G^2}{2}T^{1-2\beta} + FT^{1-\beta}\right)} \\
&= O(T^{1-\beta/2})
\end{aligned} \tag{56}$$

$\qquad\square$

It is also interesting to figure out why [13] cannot have this user-defined trade-off benefit. From [13], the key inequality in obtaining their conclusions is:

$$\begin{aligned}
\sum_{t=1}^{T} \left(f_t(x_t) - f_t(x^*)\right) &+ \sum_{i=1}^{m} \frac{\left[\sum_{t=1}^{T} g_i(x_t)\right]_+^2}{2(\sigma\eta T + m/\eta)} \\
&\leq \frac{R^2}{2\eta} + \frac{\eta T}{2}\left((m+1)G^2 + 2mD^2\right)
\end{aligned} \tag{57}$$

The main difference between Eq.(57) and Eq.(53) is in the denominator of $\frac{\left[\sum_{t=1}^{T} g_i(x_t)\right]_+^2}{2(\sigma\eta T + m/\eta)}$. Eq.(57) has the form $(\sigma\eta T + m/\eta)$, while Eq.(53) has the form $(\sigma\eta)$. The coupled $\eta$ and $1/\eta$ prevents Eq.(57) from arriving this user-defined trade-off.

The next proofs of the Proposition 4 and 5 show how we can use our proposed Lagrangian function in Eq.(7) to make the algorithms in [13] and [10] to have the clipped long-term constraint violation bounds.

*Proof of the Proposition 4.* If we look into the proof of Lemma 2 and Proposition 3 in [13], the new Lagrangian formula does not lead to any difference, which means that the $\mathcal{L}_t(x, \lambda)$ defined in Eq.(7) is also valid for the drawn conclusions. Then in the proof of Theorem 4 in [13], we can change $g_i(x_t)$ to $[g_i(x_t)]_+$. The maximization for $\lambda$ over the range $[0, +\infty)$ is also valid, since $[g_i(x_t)]_+$ automatically satisfies this requirement. Thus, the claimed bounds hold. $\qquad\square$

*Proof of the Proposition 5.* The previous augmented Lagrangian formula $\mathcal{L}_t(x, \lambda)$ used in [10] is:

$$\mathcal{L}_t(x, \lambda) = f_t(x) + \lambda g(x) - \frac{\theta_t}{2}\lambda^2 \qquad (58)$$

The Lemma 1 in [10] is the upper bound of $\mathcal{L}_t(x_t, \lambda) - \mathcal{L}_t(x_t, \lambda_t)$. The proof does not make any difference between formula (58) and (9). So we can still have the same conclusion of Lemma 1. The Lemma 2 in [10] is the lower bound of $\mathcal{L}_t(x_t, \lambda) - \mathcal{L}_t(x^*, \lambda_t)$. Since it only uses the fact that $g(x^*) \leq 0$, which is also true for $[g(x^*)]_+$, we can have the same result with $g(x_t)$ being replaced with $[g(x_t)]_+$. The Lemma 3 in [10] is free of $\mathcal{L}_t(x, \lambda)$ formula, so it is also true for the new formula. The Lemma 4 in [10] is the result of Lemma 1-3, so it is also valid if we change $g(x_t)$ to $[g(x_t)]_+$. Then the conclusion of Theorem 1 in [10] is valid for $[g(x_t)]_+$ as well. $\qquad\square$