[Reviews · NeurIPS 2018]

Reviewer 1



The authors propose algorithms for online convex optimization where the feasible solution set is speficifed by convex functions g_i's as S = {x | g_i(x) <= 0 for all i}, aiming at not only minimizing the regret, but also minimizing the cumulative violation of constraints of the form \sum_{t=1}^T ([g_i(x_t)]_+)^2 or \sum_{t=1}^T [g_i(x_t)]_+ for all i, where [a]_+ = max{0,a}. The authors claim that these results improve existing constraint-violation bounds, which bound just the sum \sum_{t=1}^T g_i(x_t). The authors also give an O(log T) regret bound with O(\sqrt{T log T}) violation when the loss functions are strongly convex. Probably, this is the first result of O(log T) regret in this setting. The result (in Section 3.2) for strongly convex loss functions are nice, although the algorithm proposed is a natural variant of Algorithm 1 with the idea from [8] taken into consideration. However, I am afraid that the results for convex loss functions (i.e., the main part of the paper) seem to be derived by the existing methods by just redefining g_i's as g_i(x) = [g_i(x)]_+, which induces the same feasible set S. Note that the function [g_i(x)]_+ is convex and L-Lipschitz whenever the function g_i(x) is. Moreover, we can further redefine g_i's as g_i(x) = c g_i(x) for a sufficently small constant c > 0, so that g_i(x) <= 1. This also preserves the convexity and the Lipschitzness. So we can assume w.l.o.g. that g_i's are defined in this way. Then, we have ([g_i(x)]_+)^2 <= g_i(x), and so the bound obtained on the squared violation \sum_t ([g_i(x)]_+)^2 is meaningless. Another concern is that it seems that the time horizon T is given to the algorithm, whereas in the previous work, T is not given. If the algorithm knows T, then we could set g_i(x) = o(1/T) by redefining g_i's as, say, g_i(x) = (1/T^2) g_i(x). So, I'm afraid that if T is known, then it would be meaningless to discuess the trade-off between the regret and the amount of constraint violation.

Reviewer 2



After rebuttal: I am now convinced that the results of the paper cannot be obtained as a direct consequence of known results. ----------- The paper studies an online convex optimization problem with long-term cumulative constraints, where the quantity representing the long-term constraint is of the form $\sum_{t=1}^T(g_i(x_t))_+^2$ instead of the basic form $\sum_{t=1}^Tg_i(x_t)$. I am sorry to say that I was not convinced by the motivation of studying such a special form of constraints: - Why not just define auxiliary constraint functions $\tilde{g}_i := g_i_+^2$ and apply known algorithms for the basic form of constraints ? Would it work? If so, how is the new algorithm better? - Why not more general form of nonnegative constraint functions $g_i$ ? A consequence of the proposed algorithm (advertised at the end of p4), is that constraint violations for each single steps $(g_i(x_t))_+$ are bounded. But I have the feeling that one could simply define auxiliary constraint functions $\tilde{g}_i := (g_i)_+$ and apply known algorithms which would provide upper bounds on Tt=1˜gi(xt)=Tt=1(gi(xt))+ which would obviously imply upper bounds on each $(g_i(x_t))_+$ ... Moreover, it seems to me that a simple way of simultaneously obtaining upper bounds on various quantities is to increase the number of constraint functions. For instance defining $\tilde{g}_i := (g_i)_+$ for $i=1,\dots, m$ and $\tilde{g}_{(i+m)} := (g_i)_+^2$ for $i=1,\dots,m$ and then applying known algorithms to get upper bounds on $\sum_{t=1}^T\tilde{g}_i(x_t)$ for $i=1,\dots,2m$.

Reviewer 3



* Summary of the paper This paper deals with the OCO framework where the learner may violate pre-defined constraints. In this setting our objective is to provide low regret with respect to the sequence of losses as well as with respect to the constraints. This setting is useful when one would like to avoid projections which might be costly. Previous works focused on the regret with the respect to the cumulative constraints, where constraint violation may be compensated by constrained satisfaction. In this work the authors consider a more appropriate measure of regret which accounts only for constraints violation. In this case, the authors come up with a new algorithm (similar in spirit to previous approaches), which provides regret guarantees with respect to the sum of squared constrained violations. They also extend their approach to the strongly convex setting. *Quality and Clarity -The paper is well written. The setting is clear and so is the algorithmic approach. Unfortunately, the authors do not provide any proof sketches in the main body of the paper, in my view this will make the paper much more complete. -I went through some of the proofs in the appendix and they look clean and sound. -Their approach does not seem to apply to the case where we have an access to noisy versions of the constraints. It will be good if the authors can discuss this issue and mention what prevents them from extending their approach to this setting. *Originality: -This paper tackles the more appropriate measure of regret for constraint violation. -Their technique of Lagrangian formulation is very similar to previous works. *Significance: In my view the main significance of this paper is in showing that one can provide guarantees with respect to the appropriate measure of constraints violation. The technique used is not very novel and is similar in spirit to other approaches. As the authors demonstrate in practice, their new approach enables to obtain better performance in several settings.

Reviewer 4



The paper studies the usual online convex optimization framework with the additional twist that the algorithm can predict a point outside of the decision set. The decision set is specified by a set of convex constraints g_i(x) <= 0, i=1,2,...,m. The goal of the algorithm is to achieve small regret with respect to the best point the decision set and at the same time achieve small sum of constraint violations. A constraint violation is penalized with the square, (g_i(x))^2. The motivation for this somewhat artificial looking problem is to avoid projections to the decision set in every round, which would be otherwise required. The paper attacks the problem via Lagrangian relaxation. The resulting algorithm is primal-dual algorithm that operates both on primal variables as well the Lagrangian variables. The resulting algorithm is an algorithm similar to subgradient descent that at the same time adjusts the Lagrangian variables. The paper proves that the algorithm has O(sqrt(T)) regret and O(sqrt(T)) sum of constraint violations in the case when both the cost functions and functions specifying the constraints are (convex) and Lipchitz. In the case when the cost functions are strongly convex and constraints are Lipchitz, the proves O(log(T)) regret bound and a bound O(sqrt(log(T) * T)) on sum of constraint violations. --- Suggestions for improvement: 1) In the introduction (Lines 29-32) g(x_t) is used without first explaining that S is expressed as S = {x : g_i(x) <= 0, i=1,2,...,m}. This is confusing and the reader is left wondering where g comes from. 2) Use a different letters of the alphabet for the unit ball \mathbb{B} and the enlarged set \mathcal{B}; different fonts are not good enough distinction. I overlooked the definition of \mathcal{B} line 70 and was left wondering if it is a typo. 3) Line 102: max --> \max 4) Line 162: argmin --> \DeclareMathOperator{\argmin}{argmin} \argmin 5) Section 5.2. You have equality constraints in equation (11). Explain how did you transform them into inequality constraints.